# Optical and Mechanical Properties of Layered Infrared Interference Filters

**DOI:** 10.3390/s22218105

**Published:** 2022-10-22

**Authors:** Michał Bembenek, Mykola Makoviichuk, Ivan Shatskyi, Liubomyr Ropyak, Igor Pritula, Leonid Gryn, Volodymyr Belyakovskyi

**Affiliations:** 1Department of Manufacturing Systems, Faculty of Mechanical Engineering and Robotics, AGH University of Science and Technology, 30-059 Kraków, Poland; 2Laboratory of Modeling of Damping Systems, Pidstryhach-Institute for Applied Problems in Mechanics and Mathematics of the National Academy of Sciences of Ukraine, Mykytynetska Str. 3, 76002 Ivano-Frankivsk, Ukraine; 3Department of Computerized Engineering, Ivano-Frankivsk National Technical University of Oil and Gas, 76019 Ivano-Frankivsk, Ukraine; 4Institute for Single Crystals, STC “Institute for Single Crystals”, National Academy of Sciences of Ukraine, 60 Nauky Ave., 61072 Kharkiv, Ukraine; 5Private company “Modern Optical Technology”, 16 Obolonskyi Ave., 04205 Kyiv, Ukraine

**Keywords:** layered coating Si–SiO, sapphire substrate, interference-absorbing filter, range of light transmission, near infrared region, local arbitrarily oriented load, stress, strength

## Abstract

The design and manufacturing technology of interference-absorbing short-wave filters based on a layered composition of Si–SiO on a sapphire substrate of various shapes was developed. A transition layer of SiO was applied to the surface of the substrate, alternating with layers of Si–SiO with an odd number of quarter-wave layers of materials with high (Si) and low refractive indices (SiO), and the application of an outer layer of SiO as an appropriate control of the materials’ thickness. The optical properties of the infrared light filter were studied. It was established that the created design of the light filter provides the minimum light transmission in the visible region of the spectrum from 0.38 to 0.78 µm and the maximum in the near infrared region from 1.25 to 5 µm and has stable optical indicators. A method for studying the stress–strain state and strength of a multilayer coating of a light filter under the action of a local arbitrarily oriented load was developed. For simplicity in the analysis and for obtaining results in the analytical form, the one-dimensional model of the configuration “multilayer covering—firm substrate” constructed earlier by authors was used. From a mechanical point of view, the upper protective layer of the multilayer coating was modeled by a flexible plate, and the inner operational composite N-layer was subjected to Winkler’s hypothesis about the proportionality of stresses and elastic displacements.

## 1. Introduction

Devices equipped with various designs of optical filters [1] for optics and spectral research are used in various fields of science and technology. For example, interference filters are used in the optical system of pyrometers. Such filters should have a high transmission coefficient in one region of the spectrum, usually in the infrared, and a high reflection coefficient in another. Infrared microscopy is rapidly developing for the study of materials, and the visualization of bacteria, tissues, etc., and for use as a quality control tool.

According to the ISO 20473:2007 standard, infrared radiation is divided into three ranges: near infrared radiation—from 0.780 to 3 μm; mid-infrared radiation—from 3 to 50 microns; far infrared radiation—from 50 to 1000 microns.

As optical elements for the near-infrared region of the spectrum, multilayer interference-absorption short-wave cut-off filters are used as part of powerful optical devices (exit window). These filters adjust the spectrum of the broadband emitter, passing infrared radiation and absorbing/reflecting visible radiation. It is known that the reflection coefficient of the light filter increases with the increase in the number of layers, and an odd number of layers exerts a greater influence on the reflective capacity of the interference coating than an even number.

The Doublet Metalens [2] was developed as a design with simultaneous chromatic and monochromatic correction in the mid-infrared range. The proposed design consists of silicon aperture metals and silicon focusing metals separated by a fused silica substrate. It is proposed to use the development in intelligent energy. The study [3] described a two-level model developed for the detection of electrical substation equipment defects from infrared images, which is based on image processing, the use of a neural network, and the construction of temperature probability density. Researchers [4] developed a SpecSeg network for specular reflection, detection, and segmentation of real-world images, which allows detection and removal of pixels that are highly affected by specular reflections.

The design of an ultra-broadband and highly efficient beam splitter based on a quasi-continuous metasurface in the near-infrared range, proposed in [5], consists of quasi-continuous triangular nanoantennas made of gallium phosphide on a silica substrate and provides a large separation angle.

The authors [6] developed an optical filter based on photonic crystals, which includes alternating layers of barium titanate ferroelectric (BaTiO_3_) and yttrium oxide (Y_2_O_3_) dielectric, with a critical defect of the high-temperature superconductor, yttrium-barium-copper oxide. It is a superconductor-based multichannel photonic crystal optical filter tunable in visible and telecommunication windows at cryogenic temperature. In [7], an integrated metalens bandpass filter based on electromagnetically induced transparency for long-wave infrared imaging for compact optical systems was developed.

The study [8] proposed a rating design consisting of a cylindrical matrix of high-performance waveguides based on graphene, which can be used for the manufacture of optical devices in the mid-infrared range (filters and refractive index sensors), and is also promising for future photonic integrated circuits. The paper [9] describes measures aimed at improving the performance of a non-dispersive infrared methane gas sensor by using multilayer thin-film bandpass filters containing germanium (Ge) and niobium pentoxide (Nb_2_O_5_) and based on alternating layers with a high and low refractive index, respectively. The researchers [10] developed a highly sensitive plasmonic layered structure containing layers of chromium, gold, chromium, and silicon dioxide deposited on a BK7 base for measuring a liquid analyte.

When designing filters, researchers mainly take into account recommendations [11,12]. They study the crystallization processes of filter lining materials [13,14,15,16], as well as the optical properties of their coatings [17].

Computer modelling is widely used to predict photovoltaic properties [18], phase composition [19,20,21] and mechanical properties of functional materials [22]. Doping with rare earth elements is used to improve the properties of optical glass [23,24]. Doping TiO_2_ oxide films with Sn improves their photocatalytic properties, and carbon with palladium nanoparticles provides it with electrocatalytic properties [25,26].

During the construction of light filters at the stage of design and technological preparation of production, it is advisable to take into account technological heredity [27] to ensure the functioning of the product throughout its life cycle [28,29]. Considerable attention is paid to thickness monitoring during the deposition of complex optical coatings [30,31,32].

Researchers [33] proposed the design of a film coating containing nano-sized aluminum fibers located at the same distance from each other. The effectiveness of the composition, evaluated by the finite element method, showed less than 26% transmission of ultraviolet and near-infrared radiation. To improve the efficiency of oxide layers, carbon nanostructures were used [34].

Wear-resistant coatings are formed by thermovacuum evaporation on piezoceramic materials and silicon probes for atomic force microscopy [35,36]. To increase the probability of trouble-free operation of infrared guidance and tracking devices, the use of electron-beam processing of optical elements was proposed [37]. The express method of determining the presence of defects on the surface of oxide coatings was developed in works [38,39].

One of the modern trends in the development of methods for assessing the strength and durability of modern structures is the improvement of interference-optical methods of the mechanics of a deformable solid body. The essence of these methods is that transparent materials become optically anisotropic under the influence of mechanical stresses [40]. The use of modern interference-absorption filters opens up new opportunities for the application of photoelasticity methods for assessing the strength and crack resistance of thin-walled structures [41] and the use of electro-optical sensors for monitoring high-precision connections and the quality of precision equipment surfaces [42,43,44,45], which operate under the influence of temperatures [46].

Calculation of thin-walled layered shells under radial load str given in works [47,48], and axial loads in [49,50]. The influence of the direction of loading and the shape of the cross-section on the Young’s modulus of the samples was established [51], as well as the influence of applications and material properties [52,53].

There are well-known methods for solving problems of thermal conductivity [54,55,56] and thermoelasticity [57,58,59] for bodies with multilayer coatings. Scientists also pay great attention to the analysis of the behaviour and reinforcing effect of thin coatings and overlays applied to cracked plates [60,61,62,63] and shells [64,65,66,67]. They studied patterns of defect development in coatings [68,69] and the technology of healing crack-like damage by injections of malleable [70,71,72] and non-contrast [73,74] fillers through surfacing [75].

As the above review shows, researchers have not paid due attention to the study of the optical and physico-mechanical properties of layered coatings under extreme conditions of high temperatures, under the action of arbitrarily oriented local mechanical loads, and the defectiveness of their structure.

This study aims to investigate the optical properties and develop an engineering method for calculating the stress state of the layered coating of the interference-absorption filter under an arbitrarily oriented local mechanical load.

To achieve the goal, the following tasks should be solved:producing experimental samples of interference light filters of various configurations;investigating their optical properties at different lengths of the light flux;developing an engineering methodology for calculating the stress state of the layered coating of the interference-absorption filter under an arbitrarily oriented local mechanical load and to investigate the stress state of the composition.

## 2. Materials and Methods

### 2.1. Materials and Manufacturing Technology of Infrared Interference Filters

Figure 1 shows the diagram of the manufacturing process of interference light filters, which includes a number of operations:

First, the workpiece of the sapphire substrate was cut and mechanical processing was carried out—diamond grinding and polishing. After that, washing, degreasing, and drying were carried out. The substrate blank prepared in this way was installed in the chamber of the device for forming light filter layers, the chamber was evacuated, and the sapphire substrate was preheated. On the flat surface of the substrate, a transition layer of SiO was deposited (by resistive method) and its thickness was controlled; a layer of Si was deposited (by electron beam method sputtered) and its thickness was controlled; a layer of SiO was then deposited and its thickness was controlled; and again a layer of Si was deposited and its thickness was controlled, etc., until the given was an odd number of light filter layers (in Figure 1 this is highlighted by a dotted line). After that, the top, outer layer of SiO was applied, its thickness was controlled, and the formation of the coating layers was completed. The installation was turned off, and after it had cooled to room temperature, the produced interference light filter was removed from the camera and its quality was controlled.

Si and SiO materials were chosen as materials with high and low refractive indices, respectively; these materials have a high affinity with each other and a large difference in refractive indices. The width of the reflection band increases as the difference in the refractive indices of the film-forming materials of the layers used increases. Also, the Si material has an absorbing capacity in the visible range of the spectrum, so the total area of high reflection increases. Layers of transparent dielectrics have great resistance to atmospheric influences and therefore they are often used as the outer layer of the filter in interference coating. The compatibility of the first matching layer with the substrate material is also of great importance for the mechanical and thermal stability of the coating, for example, according to parameters such as the coefficient of thermal expansion.

During the manufacture of experimental samples of interference light filters, a monocrystalline optical sapphire substrate was used. Monocrystalline corundum (sapphire) has a high value of thermal conductivity, while meeting one of the important requirements—effective heat removal under high temperature loads during use in powerful infrared optics. High-resistance silicon and silicon monoxide fraction 2 were used for sputtering the layers. All the materials used were particularly clean in terms of purity. Si and SiO materials have high adhesive characteristics (bond strength) and satisfactory resistance to the formation of residual thermal stresses both to the sapphire substrate and to each other, which will ensure high optical and mechanical characteristics of the wear composition and a long service life in extreme conditions.

For conducting the research, samples of interference light filters of a round shape with a diameter of 30 mm and a rectangular shape with a length of 150 mm and a width of 80 mm were made with a thickness of sapphire substrate of 3 mm. The Si–SiO interference multilayer coating was applied to a specially prepared sapphire substrate preheated to 300 °C using thermal and electron beam evaporation methods in a VU-1A vacuum unit under an excess vacuum of 3·10^−3^ Pa. The VU-1A vacuum sputtering unit was equipped with a UELI-1 electron-beam evaporation source and a resistive heater, as well as a SFKT-751 thickness photometric control complex (JSC Smorgon Optical Machine Tool Plant, Smorgon, Republic of Belarus). The SFKT-751 photometric control complex (LOMA (Leningrad Optical and Mechanical Association), St.-Petersburg, Russian Federation) was designed to control the optical thickness of films that form a coating during their application on VU-1A type vacuum units. The self-writing two-coordinate device N-307 (Zakhidprilad LLP, Lviv, Ukraine) was designed for registration in rectangular coordinates in linear or logarithmic scales of the functional dependence of two measured values, presented in the form of electrical signals of electric current voltage. The materials for sputtering the Si layer were placed in a crucible, and the SiO layer in a boat. The accelerating voltage was 12 kV, and the current was up to 250 mA. Silicon layers were sputtered by electron beam method, and silicon monoxide was deposited by resistive method. Control of the spraying process of coating layers was carried out experimentally by the photometric method using the SFKT-751 system and the N-307 device. The control wavelength was 690 nm. The measurement system consisted of the following main parts: a radiation source, a monochromator, a photoreceiver, an amplification unit, and a registration unit. The device uses the phenomenon of interference: under monochromatic illumination, the intensity of light falling on the sample increases (decreases) to ¼ the optical thickness of the film, and then begins to decrease (increase) to ½ the optical thickness. Thus, the thickness of the film can be determined by the formula: n∙d = m∙(λ/4), where n is the refractive index; d—thickness; m is the number of maxima or minima; λ is the wavelength of monochromatic light. The two-coordinate self-recording device N-307 registers extremes of light intensity.

The total number of Si–SiO layers is chosen depending on the functional purpose of the light filter. The produced interference light filter contained a transition layer with a low refractive index (SiO), an odd number of quarter-wavelength optically thick layers of materials with high (Si) and low refractive indices (SiO), the layers of which alternate, and an additional outer layer with a low refractive index (SiO), which has a larger optical thickness [76].

### 2.2. Research of Optical Properties of Infrared Interference Filters

To study the optical properties of light filters, optical transmittance studies were conducted for the ultraviolet, visible, and infrared regions of the spectrum, using round and rectangular samples.

Optical transmission tests of light filter samples were performed on:-Optizen 3220 UV (Mecasys Co., Ltd, Daejeon, Korea) double-beam spectrophotometer, which is designed for measuring transmission coefficients, optical density and scanning transmission or absorption spectra in a given wavelength range of ultraviolet and visible radiation from 190 nm to 1100 nm;-Spectrometers Spectrum One FT-IR (PerkinElmer Inc., Waltham, MA, USA) in the wavelength range from 1.25 μm to 10 μm.

Measurements were performed with a scanning step of 1 nm.

### 2.3. Mechanic-Mathematical Model of Multilayered Coating of Interference-Absorption Filter under Local Arbitrarily Oriented Loading

The structural elements of the interference-absorption filter (Figure 2) consist of the crystalline sapphire base and multi-layer film coating made of alternating Si and SiO layers with the thicknesses $h_{\mathrm{Si}}$ and $h_{\mathrm{SiO}}$, respectively, and it was determined constructively that the transitional lower $h_{tr}$—thin layer and the protective upper $h_{c}$—thick layers were to be made of SiO. This composition was loaded with inclined force P (N/m) evenly distributed along the line perpendicular to the plane of the figure.

In an effort to obtain the final result in an analytical form, in this paper we developed a one-dimensional stress analysis proposed earlier for two-layer compositions [77,78,79]. The partial case of the load perpendicular to the surface of the multilayer filter was considered by the authors earlier [80].

Mechanically, we considered the upper protective SiO-layer as a tensing and bending plate interacting with the multilayered filter. At the same time, the operational compositional N-layer Si–SiO set meets Winkler’s hypothesis on the proportionality of tangential and normal stresses to respective elastic displacements. To simplify our analysis, we assumed that the sapphire substrate was absolutely rigid, and that the mechanical contact between the components on the layer interfaces was ideal. Moreover, we assumed the plane deformation state as ($\varepsilon_{z}=0$). We analyzed the stress distribution in the layer composition of the filter structure according to the force inclination angle and specified the admissible local loading.

Taking into account the mentioned above assumptions, the equilibrium equations for the coating on the elastic substrate has the following form [81]:(1)$$B\frac{d^{2}u_{x}}{dx^{2}}-k_{x}u_{x}=-X\delta (x),\;D\frac{d^{4}u_{y}}{dx^{4}}+k_{y}u_{y}=Y\delta (x),\;x\in (-\infty ,\infty ).$$
Here $u_{x}$, $u_{y}$ are the components of elastic displacement vector of the middle plate surface; X=Pcosα, Y=Psinα; δ(x) is the Dirac function; $B=E_{c}h_{c}/(1-\nu_{c}^{2})$, $D=E_{c}h_{c}^{3}/(12(1-\nu_{c}^{2}))$ are the tension and bending rigidities; $k_{x}$, $k_{y}$ are the coefficients of integral rigidity for piecewise uniform substrate; $E_{c}=E_{\mathrm{SiO}}$, $\nu_{c}=\nu_{\mathrm{SiO}}$ are the Young’s modulus and the Poisson’s ratio of the coating material.

Let us define the coefficients of substrate rigidity for the multi-layer depth-inhomogeneous filter as the values inversely proportional to the total compliance of series-connected layers:$$k_{x}={\left[\int_{h_{c}/2}^{h_{c}/2+(N-1)(h_{\mathrm{Si}}+h_{\mathrm{SiO}})+h_{\mathrm{Si}}+h_{tr}}\frac{dy}{G(y)}\right]}^{-1}=\frac{1}{(N-1)\left(\frac{h_{\mathrm{Si}}}{G_{\mathrm{Si}}}+\frac{h_{\mathrm{SiO}}}{G_{\mathrm{SiO}}}\right)+\frac{h_{\mathrm{Si}}}{G_{\mathrm{Si}}}+\frac{h_{tr}}{G_{\mathrm{SiO}}}},\;k_{y}={\left[\int_{h_{c}/2}^{h_{c}/2+(N-1)(h_{\mathrm{Si}}+h_{\mathrm{SiO}})+h_{\mathrm{Si}}+h_{tr}}\frac{dy}{E(y)}\right]}^{-1}=\frac{1}{(N-1)\left(\frac{h_{\mathrm{Si}}}{E_{\mathrm{Si}}}+\frac{h_{\mathrm{SiO}}}{E_{\mathrm{SiO}}}\right)+\frac{h_{\mathrm{Si}}}{E_{\mathrm{Si}}}+\frac{h_{tr}}{E_{\mathrm{SiO}}}},$$
where $G_{\mathrm{Si}}=E_{\mathrm{Si}}/(2(1+\nu_{\mathrm{Si}}))$, $G_{\mathrm{SiO}}=E_{\mathrm{SiO}}/(2(1+\nu_{\mathrm{SiO}}))$ are shear moduli of Si–SiO layers.

The forces and moments vanish at infinity:(2)$$B\frac{du_{x}}{dx}(\pm \infty )=0,\;D\frac{d^{2}u_{y}}{dx^{2}}(\pm \infty )=0,\;D\frac{d^{3}u_{y}}{dx^{3}}(\pm \infty )=0.$$

Thus, the boundary problem (1), (2) describes the required field of displacements of the coating-substrate/plate on the elastic layered substrate.

## 3. Results and Discussion

### 3.1. Optical Properties of Interference-Absorption Filter

The technological features of round and rectangular light filter manufacturing are very different, especially when their geometric dimensions are increased. The asymmetry of the sample shape can affect the inhomogeneity of the functional characteristics of the product’s layered coating on the plane. Therefore, samples of light filters of various shapes and sizes were produced, and their optical and mechanical characteristics were investigated. The general appearance of light filter samples with Si–SiO interference coatings formed on round and rectangular sapphire substrates, manufactured according to the developed technology is presented in Figure 3, and the image of a round light filter sample under different types of lighting is shown in Figure 4.

The analysis of the photos (Figure 3) shows the integrity of the applied layered coating on the surface of light filters of different shapes and sizes.

A review of the pictures presented in Figure 4 shows the change in the color of the light filter sample under different types of lighting. During reflection illumination, the color of the light filter is lighter (Figure 4a) compared to the case of its translucency (Figure 4b). This means that the light filter has minimum transmittance in the visible region of the spectrum. The uniformity of illumination of the sample indicates the uniformity of the applied coating over the entire surface area. Primary visual control is necessary from the point of view of detecting inhomogeneities of characteristics over the entire surface area of the sample.

The results of the studies of the optical transmission of light filters, conducted in different ranges of radiation wavelengths from 0.19 μm to 1.1 μm and ranges from 1.25 μm to 10 μm, are presented for round samples in Figure 5 and Figure 6, and for rectangular ones in Figure 7 and Figure 8.

As we can see from Figure 5, in the spectral region from 200 to 780 nm, there is almost no transmission, and the calculation of the integral value of transmission in this region does not exceed 5%. A small peak on curves 1 and 2 in the region of 600 nm depends on the characteristics of the material and the design of the layered coating (reflectivity) and does not affect the general characteristic of the minimum light transmission in the visible region of the spectrum. Starting from 1.25 to 5 μm, the transmission of the light filter is maximum and ranges from 75 to 85% (Figure 6), which corresponds to the given optical characteristics. At the same time, good reproducibility of the test results was observed for both investigated samples 1 and 2 of the light filter. Characteristic fluctuations of optical transmittance in the infrared region of the spectrum (Figure 5 and Figure 6) are due to the interference of light passing through the light filter. The magnitude of the oscillations depends on the number of applied coating layers: the greater the number, the smaller the oscillations and the higher the values of optical transmittance in this spectrum region. Increasing the number of coating layers increases the integral reflection coefficient of the filter but does not eliminate the oscillation of the reflection coefficient in the blocking region, which leads to an increase in the integral value of the background. Also, the stress between the layers can increase, which reduces the mechanical strength of the coatings both between the substrate and the coatings and between each other. We have chosen the optimal number of coating layers to minimize transmission in the visible region and achieve the required transmission in the infrared region of the spectrum, as well as a high value of the mechanical strength of the light filter.

In order to study the optical properties of the large-sized light filter, we conducted a study of optical transmittance for different surface areas of rectangular samples in the ultraviolet, visible, and infrared regions of the spectrum. The measurement results are presented in Figure 7 and Figure 8.

The analysis of the results of the study of the transmission capacity of light filters for the ultraviolet, visible, and infrared regions of the spectrum (Figure 7 and Figure 8) shows that during the application of an interference coating on a large surface area of a sapphire substrate measuring 150 × 80 mm, a high uniformity of the optical transmission of the light filter is observed in both the ultraviolet and visible regions of the spectrum (Figure 7, surface zones 1–5), as well as in the infrared region of the spectrum (Figure 8, surface zones 1–5), i.e., for all investigated surface zones of rectangular light filter samples. Moreover, the density of lines in the infrared region of the spectrum is higher (Figure 8, surface zones 1–5). This testifies to the high uniformity of the application of the layered coating over the entire surface area of the sapphire substrate of the light filter and its high optical quality.

Since the developed light filters are proposed for use in optical pyrometers, where the entire surface is used during operation, the uniformity of optical transmission over the entire area of the sample will affect the accuracy of measuring the temperature of objects. The uniformity of the thickness of the applied coating layers on the surface of the large-sized light filter samples is ensured by the use of special equipment installed inside the chamber of the vacuum-spraying installation.

### 3.2. Stressed State of Multilayered Filter

The analytical solution to the boundary-value problem (1), (2) is built in the following form:(3)$$u_{x}(x)=\frac{X}{2B\lambda_{x}}e^{-\lambda_{x}|x|},\;\;u_{y}(x)=\frac{Y}{8D\lambda_{y}^{3}}e^{-\lambda_{y}|x|}(\cos\lambda_{y}x+\sin\lambda_{y}|x|),$$
where $\lambda_{x}=\sqrt{\frac{k_{x}}{B}}$, $\lambda_{y}=\sqrt[{4}]{\frac{k_{y}}{4D}}$ are the subgrade reaction ratios with the dimension inverse to the length.

The corresponding force and bending moment in the coating are obtained using Equation (3) for the displacements:(4)$$N(x)=B\frac{du_{x}}{dx}=\frac{X}{2}\mathrm{sgn}xe^{-\lambda_{x}|x|},\;\;M(x)=D\frac{d^{2}u_{y}}{dx^{2}}=\frac{Y}{4\lambda_{y}}e^{-\lambda_{y}|x|}(\cos\lambda_{y}x-\sin\lambda_{y}|x|).$$

The stresses in the coating are linearly distributed across the thickness



$\sigma_{x}(x,y)=\frac{N(x)}{h_{c}}+\frac{12M(x)}{h_{c}^{3}}y=\frac{X}{2h_{c}}\mathrm{sgn}xe^{-\lambda_{x}|x|}+\frac{3Y}{h_{c}^{3}\lambda_{y}}ye^{-\lambda_{y}|x|}(\cos\lambda_{y}x-\sin\lambda_{y}|x|).$



In particular at the bottom base of the coating ($y=h_{c}/2$) we obtain the relation
(5)$$\sigma_{x}(x,h_{c}/2)=\frac{X}{2h_{c}}\mathrm{sgn}xe^{-\lambda_{x}|x|}+\frac{3Y}{2h_{c}^{2}\lambda_{y}}e^{-\lambda_{y}|x|}(\cos\lambda_{y}x-\sin\lambda_{y}|x|).$$

The stresses in the filter layers have the form:(6)$$\tau_{xy}(x)=k_{x}u_{x}=\frac{X}{2}\lambda_{x}e^{-\lambda_{x}|x|},\;\sigma_{y}(x)=-k_{y}u_{y}=-\frac{Y}{2}\lambda_{y}e^{-\lambda_{y}|x|}(\cos\lambda_{y}x+\sin\lambda_{y}|x|).$$

The strength of each layer can be estimated in accordance with the von Міses criterion. Therefore, the strength condition for the plane deformed coating is as follows:(7)$$\sigma_{eq}\equiv {\left\{(1-\nu_{c}+\nu_{c}^{2})(\sigma_{x}^{2}+\sigma_{y}^{2})-(1+2\nu_{c}-2\nu_{c}^{2})\sigma_{x}\sigma_{y}+3\tau_{xy}^{2}\right\}}^{1/2}\leq {\left[\sigma \right]}_{\mathrm{SiO}},$$
and for the Winkler multi-layer substrate, it has the following form
(8)$$\sigma_{eq}^{}\equiv \sqrt{\sigma_{y}^{2}+3\tau_{xy}^{2}}\leq \min\left\{{\left[\sigma \right]}_{\mathrm{Si}},{\left[\sigma \right]}_{\mathrm{SiO}}\right\}.$$

Here ${\left[\sigma \right]}_{\mathrm{Si}},\;{\left[\sigma \right]}_{\mathrm{SiO}}$ are the admissible stresses for silicon and silicon oxide components of the filter respectively.

The obtained analytical results (4)–(8) were analyzed for the composition containing the silicon oxide coating with the parameters: $h_{c}=400$ nm, $E_{c}=E_{\mathrm{SiO}}=73$ GPa, $\nu_{c}=\nu_{\mathrm{SiO}}=0.17$, ${\left[\sigma \right]}_{\mathrm{SiO}}=110$ MPa; the silicon oxide transition layer $h_{tr}=10$ nm; the silicon layer with the parameters: $h_{\mathrm{Si}}=50$ nm, $E_{\mathrm{Si}}=131$ GPa, $\nu_{\mathrm{Si}}=0.266$, ${\left[\sigma \right]}_{\mathrm{Si}}=130$ MPa; the silicon oxide layer with the parameters: $h_{\mathrm{SiO}}=100$ nm, $E_{\mathrm{SiO}}=73$ GPa, $\nu_{\mathrm{SiO}}=0.17$, ${\left[\sigma \right]}_{\mathrm{SiO}}=110$ MPa. The number of bi-layers was N=6.

Figure 9, Figure 10 and Figure 11 show the stressed state of the composition.

As we can see on represented graphs, the most dangerous place is the point x=0, y=h/2 in coating for the all-load directions.

The admissible loadings $P_{*}$ were found from the conditions (7), (8) taking into account, that $\underset{x}{\max}\sigma_{eq}(x)=\sigma_{eq}(0)$. In particular, as Figure 11 shows, the most dangerous situation is observed for perpendicular loading (α = 90°) and for the coating $\sigma_{eq}(0)\approx 1.65\;P/h_{c}$. In this case, the maximum permissible distributed load along the loading line should have the value

$P_{*}=\frac{{\left[\sigma \right]}_{\mathrm{SiO}}h_{c}}{1.67}=$ $\frac{110\cdot 10^{6}\cdot 400\cdot 10^{-9}}{1.65}=26.67$ N/m.

Taking into account that the theoretical strength of SiO-nano-layers is bigger than ${\left[\sigma \right]}_{\mathrm{SiO}}=110$ MPa, we can essentially increase the obtained value of the admissible load.

## 4. Conclusions

The design of an interference-absorbing short-wave filter based on a layered composition of Si–SiO on a sapphire substrate has been developed. The light filter manufacturing technology includes the application of a transition layer of SiO to the surface of the substrate, alternating Si–SiO layers (an odd number of quarter-wavelength layers of materials with high (Si), and low refractive indices (SiO), alternating layers), completing the process of forming the light filter coating with the application of an outer SiO layer with appropriate control of the layers’ thickness.

Based on the results of the optical properties study, it was established that the light filter provides minimum light transmission in the visible part of the spectrum from 0.38 to 0.78 µm and maximum light transmission in the near-infrared region from 1.25 to 5 µm and has stable optical indicators over the entire surface area of the samples.

The main feature of the proposed technique of mechanical analysis is the use of strength criteria for all components of a partially homogeneous layered structure. For the layered Si–SiO composition on a sapphire substrate, the stress–strain state and the allowable load were evaluated and their dependence on the angle of inclination of the applied load was analyzed. The calculation method proposed by the authors will allow engineers to control, in the analytical form, the influence of the ratio of geometric and mechanical characteristics of nanolayers on the stress state and the limit equilibrium of the interference-absorbing filter depending on the magnitude and orientation of the localized load.

Studies have shown that the most dangerous stresses are in the coating, which should be taken in account in the strength calculation, and the worst force inclination angle is the case of perpendicular loading.

In the future, the effect of heating on the change in the stress–strain state of the layered light filter will be investigated.

## Figures and Tables

**Figure 1 sensors-22-08105-f001:**
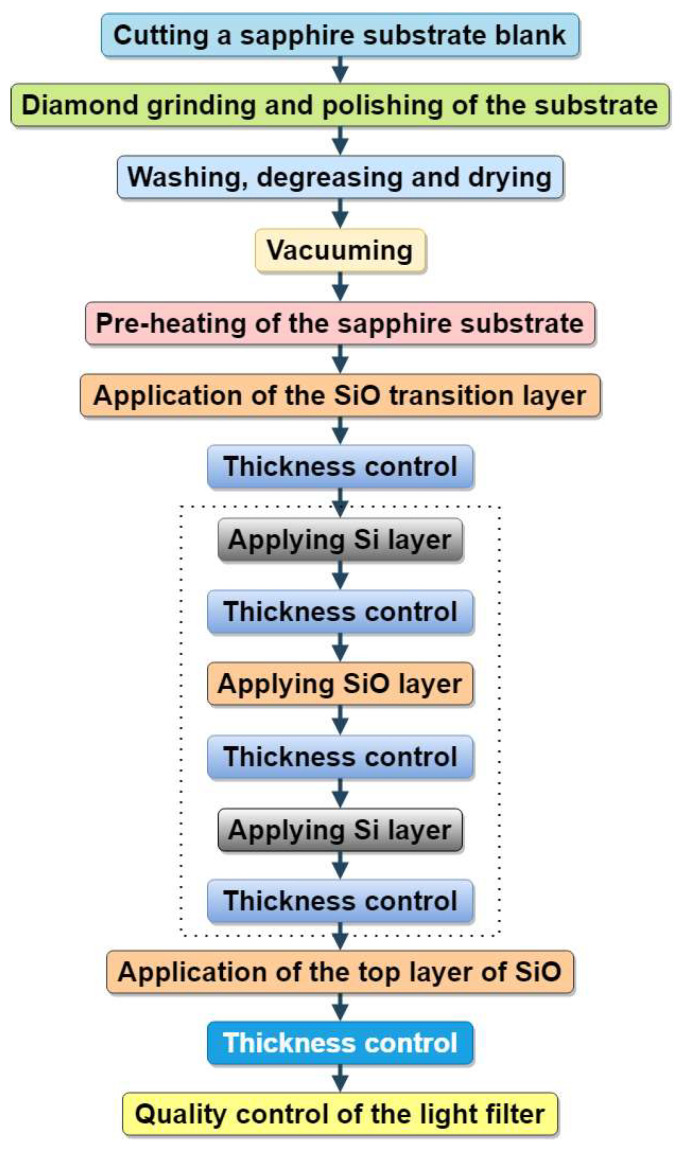
Block diagram of the technological process of forming infrared interference filters.

**Figure 2 sensors-22-08105-f002:**
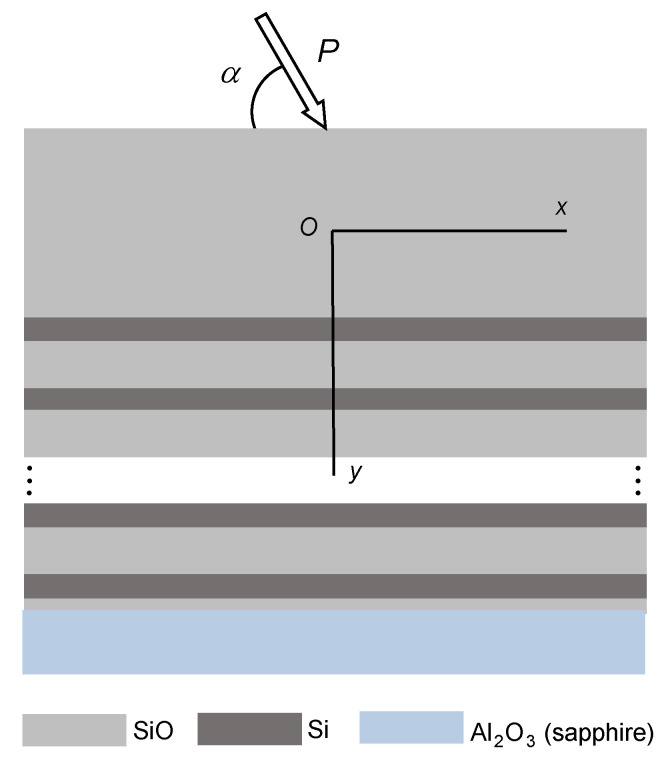
A fragment of the laminated filter under inclined loading.

**Figure 3 sensors-22-08105-f003:**
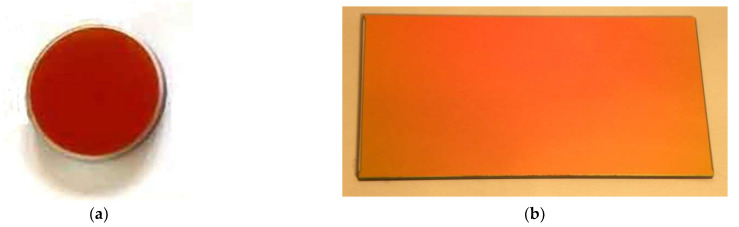
The general view of the optic filters with interference Si–SiO coating, formed on a sapphire substrate: (**a**)—a round sample with a diameter of 30 mm; and (**b**)—a rectangular sample with dimensions of 150 × 80 mm.

**Figure 4 sensors-22-08105-f004:**
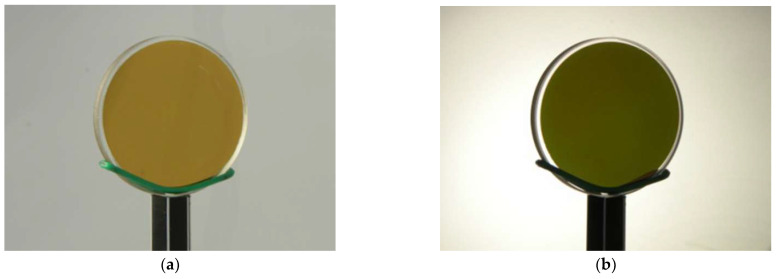
The general view of the round optic filter with a diameter of 30 mm with interference Si–SiO coating at a different lighting: (**a**)—illumination for reflection; and (**b**)—enlightenment.

**Figure 5 sensors-22-08105-f005:**
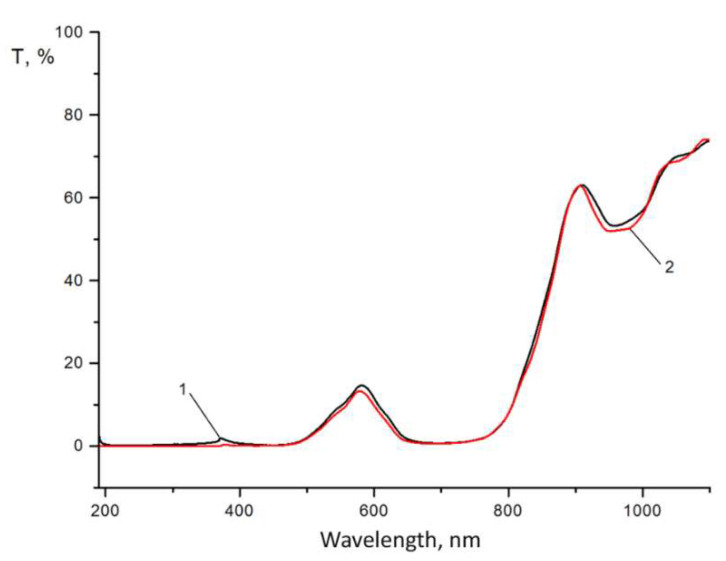
Optical transmittance in the ultraviolet and visible regions of the spectrum of rounded samples 1 and 2 with interference Si–SiO coating, formed on a sapphire substrate with a diameter of 30 mm.

**Figure 6 sensors-22-08105-f006:**
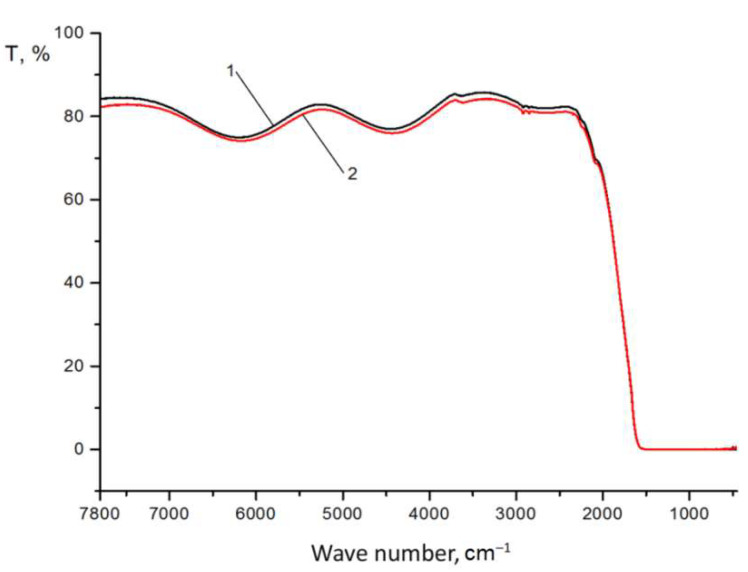
Optical transmittance in the infrared region of the spectrum of rounded samples 1 and 2 with interference Si–SiO coating, formed on a sapphire substrate with a diameter of 30 mm.

**Figure 7 sensors-22-08105-f007:**
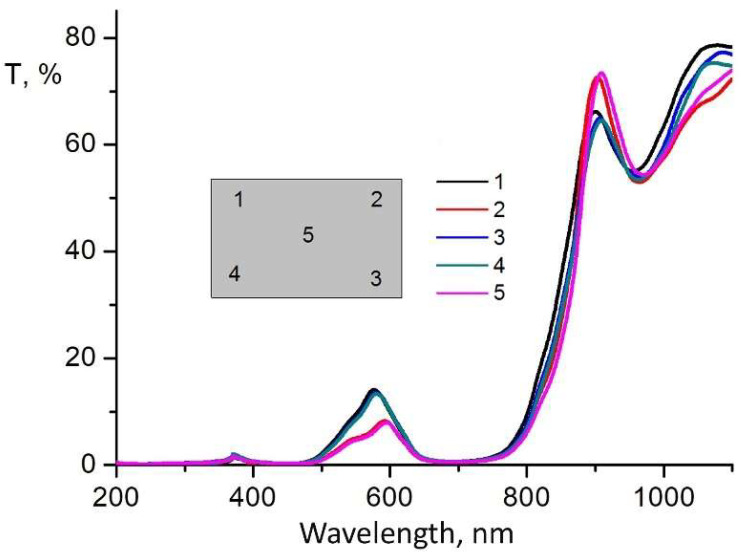
Optical transmittance in the ultraviolet and visible regions of the spectrum of samples with interference Si–SiO coating, formed on a sapphire substrate with dimensions of 150 × 80 mm at the investigated 1–5 zones.

**Figure 8 sensors-22-08105-f008:**
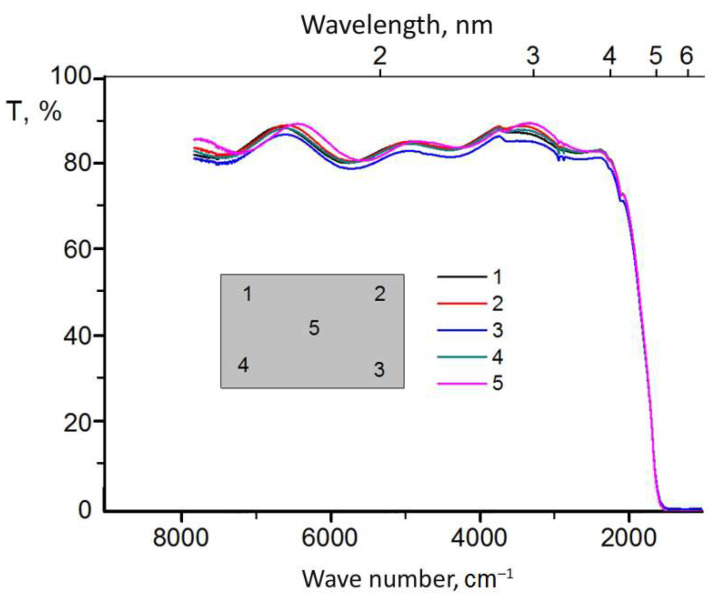
Optical transmittance in the infrared region of the spectrum of samples with interference Si–SiO coating, formed on a sapphire substrate with dimensions of 150 × 80 mm at the investigated 1–5 zones.

**Figure 9 sensors-22-08105-f009:**
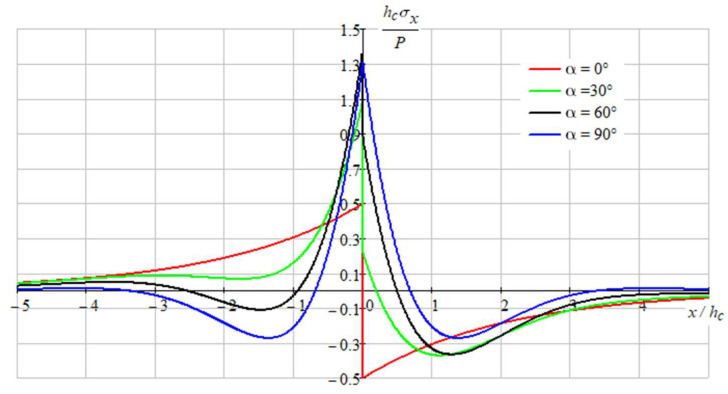
Distribution of normal stresses $\sigma_{x}$ in the coating at the boundary of contacts between the layers ($y=h_{c}/2$).

**Figure 10 sensors-22-08105-f010:**
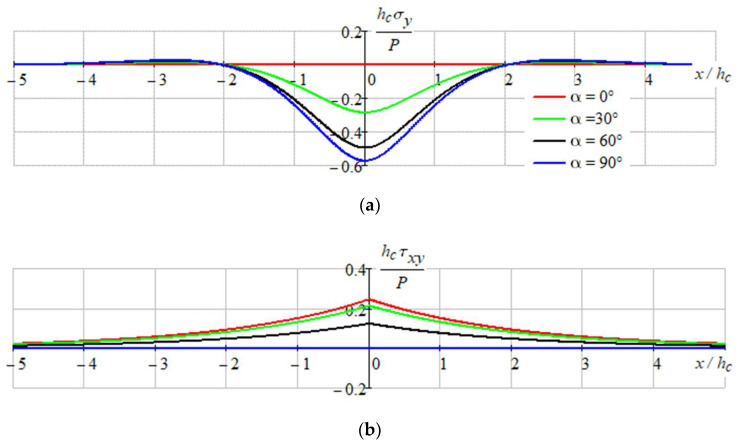
Distribution of stresses: (**a**)—$\sigma_{y}$; and (**b**)—$\tau_{xy}$ in the filter layers.

**Figure 11 sensors-22-08105-f011:**
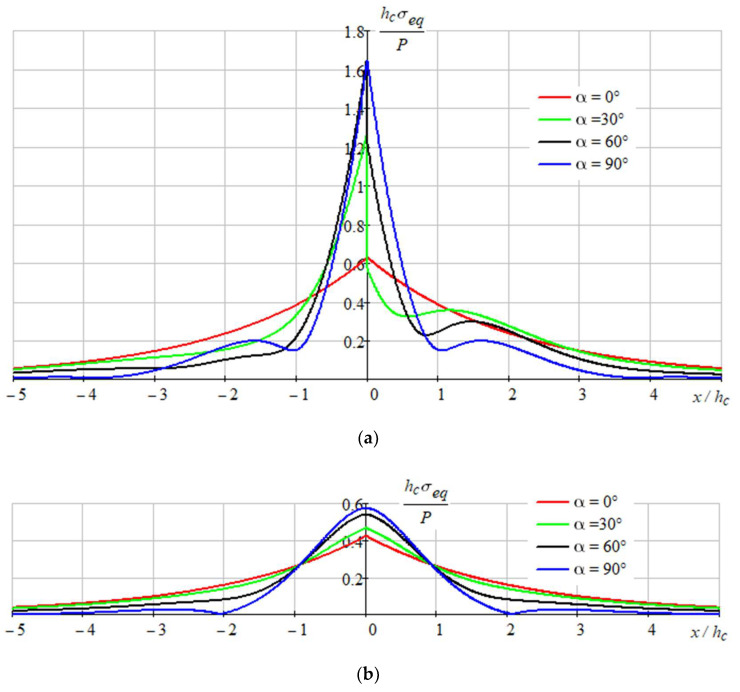
Distribution of equivalent stresses: (**a**)—in the coating at $y=h_{c}/2$; and (**b**)—in the filter.

## Data Availability

Data are contained within the article.

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
