# Peer review of "Optical and Mechanical Properties of Layered Infrared Interference Filters"

_sensors, 2022, doi:10.3390/s22218105_

Round 1

Reviewer 1 Report

1. In terms of content, the selection of materials is not relatively novel. Si and SiO are common thin film materials. The production process is also a conventional resistance evaporation and electron beam evaporation process. The design of the film system is a standard optical thickness design. The only novel one is the one-dimensional model of "multilayer covering solid substrate" proposed by the author, which is mainly the mechanical properties of the material structure, independent of the optical properties. In order to improve the quality of the article, please add more cutting-edge innovations to the optical characteristics of the filter.

2. From the perspective of language, the overall language expression of the article needs to be improved, which can be modified by English speaking authors.

3. In terms of structure, the overall structure of the article is clear, the introduction part is slightly lengthy, and the conclusion part needs to add more specific data and theoretical analysis. The current content is closer to summary than conclusion.

4. The first part: Abstract. The abstract part is based on the standard of concise and comprehensive content. The content of this article is comprehensive, but the number of words is too large to be refined.

5. The second part: Introduction, has made an effective analysis of the existing technology, and pointed out the problems of the advanced research content and the research direction of this paper. The content is complete and clear.

6. The third part: Materials and Methods. This part does not need too much description of detailed operations. It mainly focuses on the analysis and understanding of material selection, as well as the introduction of implementation methods and the derivation of theoretical formulas. The title of this paper is the study of optical properties and mechanical properties, but the actual content of optical properties covers less and lacks theoretical analysis.

7. The fourth part: Results and discussion. The description of the graph of the article needs to be surrounded by process, and the horizontal line of the countermeasure can more clearly show the trend of the curve. All test contents shall indicate the model, manufacturer, country and specific parameters of the equipment.

8. Part V: Conclusions. As mentioned above, the conclusion tends to be summary. And it is necessary to increase the prospect of more advanced technology and the future scientific research direction.

9. In addition, all the figure descriptions are not specific enough. Please add more content to the picture descriptions so that readers can get complete information from the data.

Author Response

The answers are in the attached file.

Reviewer 2 Report

Overall, this manuscript needs an extensive English revision. There are a lot of spelling and graphical errors (e. g. “with ] with” (line 59); sm-1 (xx axis of figure 6); abstract has different font size).

It is difficult to understand the main innovations of this work, probably because the paper is difficult to read.

In section 2, materials and methods, some relevant information is missing, e. g.

- How is performed the thin-film thickness control, inside the deposition chamber?

- Equipment used – profilometer, ellipsometer?

In section 3, the study presented in Figures 7 and 8 should be done with the same zones 1-5. The uniformity will be hampered by the deposition camera, I mean its uniformity of the layer thickness. This should be discussed.

Author Response

The answers are in the attached file.

Round 2

Reviewer 1 Report

The authors corrected the errors in the article and finalized the article in accordance with my comments.

Reviewer 2 Report

The authors replied to all my questions and made the necessary corrections, I believe the paper was improved. Some more English revisions can be performed before publication.